# Ischiofemoral Impingement Syndrome: Clinical and Imaging/Guidance Issues with Special Focus on Ultrasonography

**DOI:** 10.3390/diagnostics13010139

**Published:** 2022-12-31

**Authors:** Wei-Ting Wu, Ke-Vin Chang, Kamal Mezian, Ondřej Naňka, Vincenzo Ricci, Hsiang-Chi Chang, Bow Wang, Chen-Yu Hung, Levent Özçakar

**Affiliations:** 1Department of Physical Medicine and Rehabilitation, National Taiwan University Hospital, Bei-Hu Branch, Taipei 10845, Taiwan; 2Department of Physical Medicine and Rehabilitation, College of Medicine, National Taiwan University, Taipei 10048, Taiwan; 3Center for Regional Anesthesia and Pain Medicine, Wang-Fang Hospital, Taipei Medical University, Taipei 11600, Taiwan; 4Department of Rehabilitation Medicine, First Faculty of Medicine and General University Hospital in Prague, Charles University, 12800 Prague, Czech Republic; 5First Faculty of Medicine, Institute of Anatomy, Charles University, 12800 Prague, Czech Republic; 6Physical and Rehabilitation Medicine Unit, Luigi Sacco University Hospital, ASST Fatebenefratelli-Sacco, 20157 Milan, Italy; 7Department of Physical Medicine and Rehabilitation, Taichung Veterans General Hospital, Taichung 40705, Taiwan; 8Department of Medical Imaging, National Cheng Kung University Hospital, College of Medicine, National Cheng Kung University, No. 1 University Rd., Tainan 70101, Taiwan; 9Department of Physical and Rehabilitation Medicine, Hacettepe University Medical School, Ankara 06100, Turkey

**Keywords:** ischiofemoral impingement, hip pain, quadratus femoris, magnetic resonance imaging, ultrasound

## Abstract

Ischiofemoral impingement syndrome is a neglected cause of posterior hip pain which is derived from narrowing of the space between the lateral aspect of the ischium and the medial aspect of the lesser trochanter. Its diagnosis is challenging and requires the combination of physical tests and imaging studies. In the present narrative review, we found that femoral anteversion predisposes patients to the narrowing of the ischiofemoral space and subsequent quadratus femoris muscle injury. Magnetic resonance imaging serves as the gold-standard diagnostic tool, which facilities the quantification of the ischiofemoral distance and the recognition of edema/fat infiltration/tearing of the quadratus femoris muscle. Ultrasound is useful for scrutinizing the integrity of deep gluteal muscles, and its capability to measure the ischiofemoral space is comparable to that of magnetic resonance. Various injection regimens can be applied to treat ischiofemoral impingement syndrome under ultrasound guidance and they appear to be safe and effective. Finally, more randomized controlled trials are needed to build solid bases of evidence on ultrasound-guided interventions in the management of ischiofemoral impingement syndrome.

## 1. Introduction

A population-based survey investigating 2600 adult participants aged between 38 and 77 years reported that the prevalence of self-reported hip disorders was 32% [1]. Its prevalence increased with age and reached 42% among females aged between 48 and 67 years. Regarding its clinical manifestations, pain was the most common symptom (86%), followed by stiffness (32%) and weakness (20%).

The posterior hip is an important region, where pain can ensue due to piriformis syndrome, osteoarthritis and sacroiliac joint disorders. Ischiofemoral impingement syndrome (IIS) is a neglected cause of posterior hip pain which originates from the narrowing of the space between the lateral aspect of the ischium and the medial aspect of the lesser trochanter [2]. The syndrome is commonly associated with an antecedent trauma and/or hip surgery [3]. Its diagnosis is challenging and requires the combination of physical tests and imaging studies. In recent years, ultrasound (US) has emerged as one of the most useful tools to assess musculoskeletal disorders [4], including those of the hip joint [5,6]. Likewise, in the present narrative review, special focus is given to US examinations and guided interventions for IIS, in addition to providing a comprehensive overview on its clinical aspects.

## 2. Anatomy

The ischiofemoral space is demarcated by the ischial tuberosity medially and the lesser trochanter of the femur laterally [7]. The cranial end is bordered by the femoral neck, ischiofemoral ligament and inferior gemellus muscle. The caudal end is defined by the inferior edge of the quadratus femoris, a quadrangular muscle originating from the ischial tuberosity attached to the intertrochanteric crest of the femur (Figure 1). The quadratus femoris muscle is innervated by the nerve to the quadratus femoris, which receives fibers from nerve roots L4 to S1, and is supplied by the inferior gluteal artery [8].

Deep in the quadratus femoris is the obturator externus muscle, originating from the obturator membrane and bony boundaries of the obturator foramen and being inserted into the trochanteric fossa of the femur (Figure 2) [9]. The obturator externus is innervated by the obturator nerve, derived from the L3 and L4 nerve roots, and is supplied by the obturator and medial femoral circumflex arteries. The two aforementioned muscles are commonly involved in IIS. It is noteworthy that the conjoint tendon of the semitendinosus, semimembranosus and biceps femoris muscles have proximal attachments on the ischial tuberosity [10], and their pathologies are scarcely differentiated from IIS. Moreover, sciatic and inferior gluteal nerves course through the ischiofemoral space, and their entrapment may lead to symptoms of sciatica.

## 3. Biomechanics

The configuration of the femoral neck in relation to the femoral shaft plays a role in the alignment of the lesser trochanter against the ischial tuberosity and it also influences the ischiofemoral space. The angle between the femoral neck and shaft is termed the femoral neck anteversion angle, which is associated with the magnitude of femoral torsion [11]. The femoral neck anteversion angle is commonly measured in the axial plane. A line is first drawn from the center of the femoral head to the midpoint of the greater trochanter to denote the axis of the femoral neck. A line is subsequently depicted along the posterior edge of the medial and lateral femoral condylar cartilage to represent the axis of the femoral shaft. The angle between the two aforementioned lines is equal to the femoral neck anteversion angle.

The range of the normal femoral anteversion angle is between 15 and 20 degrees in adults, although it can reach 40 degrees at birth [12]. Participants with excessive femoral anteversion have a compensatory toe-in posture during ambulation to let the center of the femoral head approximate the center of the acetabulum. On the other hand, people with a femoral anteversion angle of less than 14 degrees, also known as femoral retroversion [13], tend to compensate by using a toe-out posture while walking (Figure 3).

In the toe-in posture, the femur is internally rotated and this tightens the deep gluteal muscles, including the quadratus femoris and obturator externus. It is well known that an overstretched muscle is vulnerable to strain injury [14]. Furthermore, femoral anteversion causes the lesser trochanter to pivot backward, leading to encroachment of the quadratus femoris muscle. In this sense, excessive femoral anteversion serves as a risk factor for IIS. The aforementioned inference has been proven by a magnetic resonance imaging (MRI) study [15], which showed that the femoral neck anteversion angle was significantly higher in patients with IIS than asymptomatic controls.

## 4. Physical Examination

Two physical tests have been commonly performed for the diagnosis of IIS [16]. The long-stride walking test (Figure 4) is conducted by asking the patient to grab the buttock lateral to the ischium and extend the affected hip. If pain is elicited, the diagnosis is likely. Herewith, it should be confirmed by the absence of pain during walking with short strides. Using MRI and computed tomography (CT) of the ischiofemoral space as the gold standard, the sensitivity and specificity of the long-stride walking test have been reported as 0.94 and 0.85, respectively [16].

The ischiofemoral impingement test is conducted using the lateral decubitus position with the affected side facing upward. The examiner presses the buttock slightly lateral to the ischium after passively extending the hip. If pain is provoked at the neutral/adducted (but not abducted) femur, IIS is likely (Figure 5). Using MRI or CT measurements of the ischiofemoral space as the reference, the sensitivity and specificity of the ischiofemoral impingement test were reported as 0.82 and 0.85, respectively [16].

Recently, a two-step provocation test has been proposed [17]. In the first step, the participant stands with one hand supporting them against the wall while the other hand grasps a non-weight bearing foot, and flexes the ipsilateral knee and extends the ipsilateral hip. In the second step, the participant is in the lateral decubitus position with the asymptomatic side touching the bed. The examiner uses one hand to flex the knee on the symptomatic side and externally rotate the ipsilateral hip after the pelvis is stabilized using the other hand. If buttock pain is elicited in either one of the two steps, the patient is assumed to have IIS (Figure 6). An MRI study reported that pain provoked by the aforementioned two-step test was associated with decreased ischiofemoral width [17].

There are also some nonspecific physical tests for the diagnosis of IIS which might be helpful for differentiating the origin of gluteal pain. For instance, during the FADIR (flexion, adduction and internal rotation) test [18], the patient is positioned supine with the target hip and ipsilateral knee flexed. The examiner grasps the ipsilateral leg, followed by adduction and internal rotation of the hip. If pain develops over the buttock, involvement of the piriformis, deep gluteal muscles and posterior hip capsule should be taken into account. If pain is elicited over the groin, lesions over the anterior hip joint and iliopsoas tendon should be considered.

## 5. Clinical Manifestations

Patients with IIS typically have pain over the posterior buttock, aggravated by a long duration of weight bearing or hip extension/adduction [2]. Snapping or clunking might be felt over the posterior hip joint. In cases with concomitant sciatic nerve entrapment, patients have pain and sensory disturbance (numbness, tingling, prickling, etc.) radiating to the ipsilateral knee, leg and foot. Due to limited hip extension, patients might develop associated lower-back pain caused by increased pressure on the lumbar facet joints [19]. A positive Trendelenburg position could be expected in those with accompanying weakness of the gluteus medius muscle.

## 6. Differential Diagnosis

There are miscellaneous disorders mimicking IIS, e.g., lumbosacral radiculopathy, sacroiliac joint dysfunction, sciatic nerve entrapment, posterior hip joint arthropathy and ischiofemoral ligament sprain. Considering the overlapping clinical presentations and anatomy related to IIS, deep gluteal and hamstring syndromes seem to be the two scenarios that should be differentiated.

Deep gluteal syndrome can be defined as non-discogenic posterior buttock pain with possible irritation of the sciatic nerve [20]. It is likely to be of muscular origin, with muscles in proximity to the greater sciatic notch (e.g., the posterior band of the gluteus medius, gluteus minimus, piriformis, superior gemellus and inferior gemellus muscles) are involved. Palpation is crucial for its diagnosis and this should be initiated at the level of the posterior iliac wing, followed by caudal relocation to the greater sciatic notch. Flexion, adduction and internal rotation of the hip joint can serve as a provocation test, with stretching the piriformis and the relevant hip external rotator muscles.

On the other hand, hamstring syndrome is manifested by pain on or adjacent to the ischial tuberosity, radiating to the posterior thigh [21]. It is frequently elicited by prolonged sitting. Pain can originate (proximally) from the tendinous insertion of the hamstring muscles on the ischial tuberosity distal to its myotendinous junction. Likewise, palpation of the aforementioned area is helpful for establishing the diagnosis. An active knee extension test can be applied during examination of the hamstring muscle tightness. The patient lies supine with the target hip flexed and the ipsilateral knee extended. A lag of more than 20 degrees from full knee extension suggests hamstring tightness.

## 7. MRI Assessment

MRI is the imaging tool most commonly used for the assessment of IIS. The width of the ischiofemoral space and the thickness of the quadratus femoris muscle can be estimated using axial views. Compared with T2 weighted imaging, T1 sequences are better for these measurements owing to their superiority in delineating regional anatomy. The smallest distance between the lateral cortex of the ischial tuberosity and the medial cortex of the lesser trochanter is used to represent the ischiofemoral space (Figure 7). On the other hand, the quadratus femoris space, which is bordered by the superolateral surface of the hamstring tendons and the attachment of the iliopsoas tendon (or the lesser trochanter), is considered to be the narrowest area for the quadratus femoris muscle to pass [22]. A meta-analysis of four studies revealed that the average ischiofemoral and quadratus femoris spaces were significantly smaller in symptomatic patients than asymptomatic controls [23]. Using a cut-off value of 15 mm for the ischiofemoral space; the sensitivity, specificity and overall accuracy were 76.9%, 81.0% and 78.3%, respectively. Furthermore, if the threshold of the quadratus femoris space was set at 10 mm; the sensitivity, specificity and overall accuracy were 78.7%, 74.1% and 77.1%, respectively.

In addition to the quantitative evaluation of the ischiofemoral space, MRI is useful in the qualitative assessment of the quadratus femoris muscle (Figure 8 and Figure 9). At the initial stage, the muscle might become edematous, i.e., presenting with hyperintensity on T2 weighted images. A grading system for quadratus femoris muscle edema was proposed in 2012: (1) Grade I, normal; (2) Grade II, focal edema; (3) Grade III, diffuse edema; and (4) Grade IV, edema extending to the surrounding tissues outside the muscle [24]. At the chronic stage, the quadratus femoris muscle becomes atrophic by being replaced by adipose tissue. The following grading system was introduced as regards fatty infiltration of the quadratus femoris muscle using T1 weighted imaging: (1) no increase in signal intensity, (2) tiny and linear increases in signal intensity, (3) thicker and linear-globular increases in signal intensity, involving less than 50% of the muscle and (4) globular increases in signal intensity, involving more than 50% of the muscle [24]. In severe cases, the quadratus femoris can be partially or completely ruptured [25]. Hydroxyapatite crystal deposition in the muscle, shown with a low intensity in T1-weighted imaging, can also result in IIS [26]. Other reported causes include chronic avulsion injury of the hamstring tendon [27] and osteochondroma of the lesser trochanter [28].

## 8. Dynamic Ultrasound Evaluation

Before the assessment of structures inside the ischiofemoral space, the examiner must be aware of the explicit evaluation of the hip external rotators [29]. The curvilinear transducer would be preferable to image the deep structures. The transducer can be placed in the horizontal plane at the posterior superior iliac spine level, revealing the gluteus maximus, gluteus medius and gluteus minimus muscles from the superficial to the posterior iliac fossa. The transducer is then moved more caudally until the greater sciatic notch, the bony discontinuity over the posterior iliac wall, is visualized. Pointing the transducer to the greater trochanter, the piriformis muscle can be seen emerging from the anterior surface of the sacrum, passing through the greater sciatic notch and next to the concave portion of the ischium, attaching to the trochanteric fossa [30]. Adjusting the transducer back to the horizontal plane and moving it caudally, the superior gemellus muscle can be recognized, extending from the ischial spine (with a long straight bony cortex) to the trochanteric fossa (Figure 10).

More distally, the obturator internus can be seen originating from the inner surface of the obturator foramen, wrapping around the ischium and attaching to the trochanteric fossa. The presence of an intra-muscular tendon and a ‘waterfall moving pattern’ upon external/internal femoral rotation indicates the obturator internus. Relocating the transducer on top of the posterior hip joint, the inferior gemellus appears over the ischiofemoral ligament. Lastly, the quadratus femoris arises from the deeper portion of the ischium and the inferior edge of the semimembranosus and is inserted on the intertrochanteric crest. The muscle situated underneath the quadratus femoris is the obturator externus (Figure 11).

Another clinical issue is whether the ischiofemoral space can be easily/reliably measured by using US (Figure 12). In 2019, a 15-min educational course with 20 min of hands-on training was given to nine physician sonographers [31]. Subsequently, each examiner conducted US measurements of the ischiofemoral space twice. The intra-rater reliability, quantified by interclass coefficients, ranged between 0.596 and 0829, whereas the inter-rater reliability ranged between 0.427 and 0.722.

In 2017, the capability of US imaging for measuring the ischiofemoral space was validated via MRI on 10 asymptomatic participants [32]. The examiner toggled the transducer cephalad and caudad in the axial plane using the heel-toe maneuver to clearly visualize the ischiofemoral space. The measurement was implemented between the medial cortex of the lesser trochanter and the lateral cortex of the ischial tuberosity. The difference between MRI and US measurements was insignificant (mean value: 1.25 mm; 95% confidence interval, −0.49 mm to 2.98 mm). The US-measured distance was positively correlated with that obtained via MRI (*r* = 0.781, *p* < 0.001).

In 2019, another validation study was conducted on 16 patients with hip pain and 19 controls [33]. The ischiofemoral space measured via US was significantly shorter in patients than controls. US measurements of the ischiofemoral space exhibited positive correlations with MRI (*r* = 0.575, *p* = 0.01) in both patient and control groups. Using the cut-off value of 2.14 cm, US exhibited a sensitivity of 92.0% and a specificity of 68.4% for the detection of IIS.

US is helpful in delineating the configuration of the sciatic nerve, which is strongly implicated in IIS. In 2018, sonographic examinations were conducted on six fresh cadavers and 31 healthy volunteers [34]. The magnitude of the sciatic nerve excursion appeared to be more noticeable during hip external/internal rotation than in the neutral position. Furthermore, sciatic nerves were found to change their alignment from a straight to a curved shape following hip internal rotation, which increases the risk of nerve tethering (Figure 13).

## 9. US-Guided Intervention

A stepwise algorithm was published in 2017 for the management of IIS [35]. In symptomatic patients, plain radiography, CT or MRI can be first used to scrutinize the presence of tumors or exostoses for which surgical excision should be performed. Otherwise, exercises such as quadratus femoris stretching and strengthening can be prescribed. If conservative treatments fail to achieve satisfactory effects, imaging-directed interventions should be taken into account. Although computed tomography can be used for guidance, due to the concern of radiation exposure, US guidance is the preferred choice.

In 2013, Volokhina et al. demonstrated a case with posterior hip pain and quadratus femoris muscle edema on axial T1 and proton-density fat-suppressed MRI [36]. A combination of 3 mL lidocaine and 40 mg methylprednisolone was given into the ischiofemoral space, using in-plane US guidance, by piercing the proximal hamstring tendon. In 2014, Backer et al. conducted a retrospective study on seven patients undergoing US-guided quadratus femoris muscle injections with 1 mL of triamcinolone and 3 mL of 1% lidocaine [37]. The needle was introduced from the lateral to the medial direction in order to avoid sciatic nerve damage (Figure 14). An average reduction of 1.7 (over a 10 cm visual analogue scale of pain) was documented two weeks after injection. In 2014, Kim et al. demonstrated the feasibility of US-guided prolotherapy for the treatment of IIS [38]. The out-of-plane approach was carried out medially to the sciatic nerve using polydeoxyribonucleotide sodium (3 mL) mixed with mepivacaine (1 mL) on two cases, which was proven to be effective even six months after the injection. In 2016, Kim et al. conducted a retrospective evaluation of 14 patients receiving US-guided quadratus femoris muscle injection with 8 mL of 0.25% lidocaine [39]. Two weeks after the injections, a mean decrease of 49.3% in pain and 70% satisfaction in treatment outcome were reported. In 2018, Chen et al. demonstrated upward displacement of the quadratus femoris muscle during external hip rotation (‘the eruption sign’) in a 34-year-old female patient [40]. Two sessions of 100 IU botulinum toxin A were administered into the quadratus femoris muscle under in-plane US guidance. On the 5th month post-injection, the patient successfully returned to her premorbid activity level and follow-up US imaging revealed thinning of the quadratus femoris muscle with the absence of the eruption sign.

In light of our literature search, we observed that the current evidence regarding US-guided injections for IIS was limited, i.e., comprising case reports/series. Further randomized controlled trials comparing US-guided interventions with other non-operative approaches (e.g., physical therapy and exercises) are indisputably needed.

## 10. Conclusions

IIS is an overlooked entity within the spectrum of posterior hip pain. It usually presents with symptoms that are not distinguished from deep gluteal and hamstring syndromes. Femoral anteversion predisposes patients to narrowing of the ischiofemoral space and subsequent quadratus femoris muscle injury. MRI serves as the gold-standard diagnostic tool, which facilitates the quantification of the ischiofemoral distance and the recognition of edema, fat infiltration and tearing of the quadratus femoris muscle. US is useful for scrutinizing the integrity of deep gluteal muscles and its capability to measure the ischiofemoral space is notably comparable to that of MRI. Various injectates have been applied to treat IIS under US guidance and they appear to be safe/effective. Finally, future randomized controlled trials are needed to solidify the evidence regarding US-guided interventions in the management of IIS.

## Figures and Tables

**Figure 1 diagnostics-13-00139-f001:**
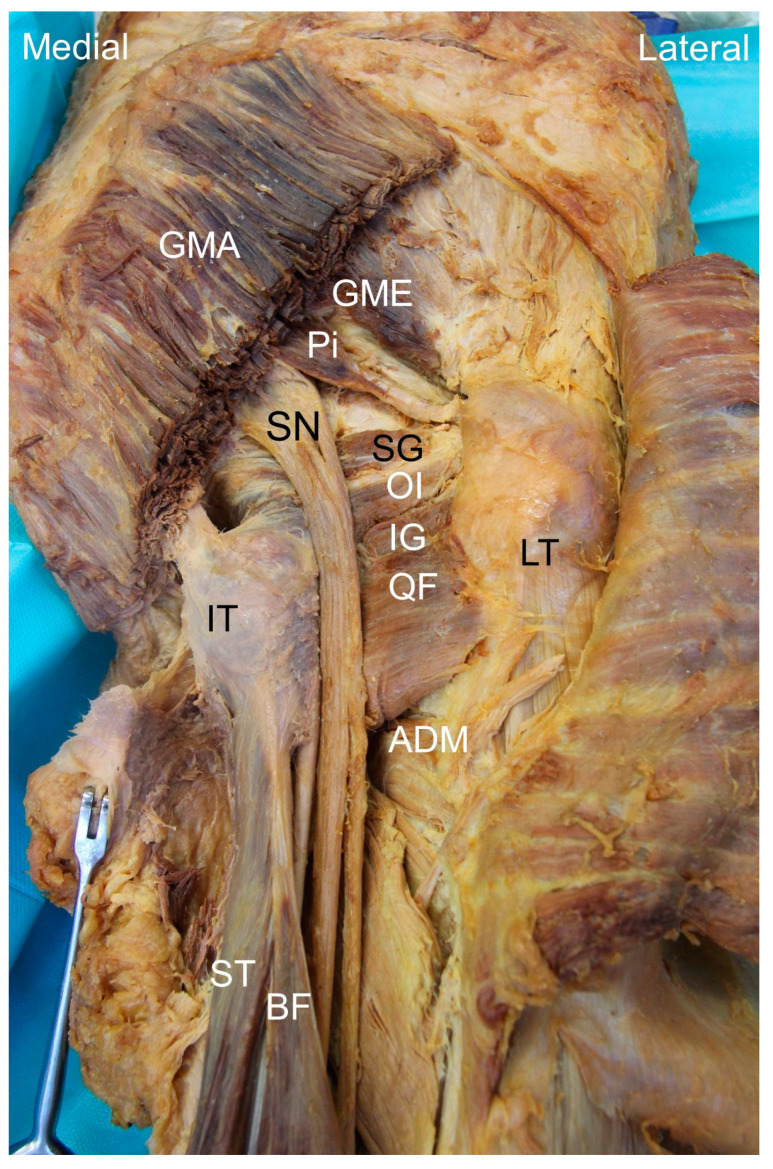
Cadaveric image of the posterior hip region, focusing on the ischiofemoral space. GMA: gluteus maximus muscle; GME: gluteus medius muscle; Pi: piriformis muscle; SG: superior gemellus muscle; OI: obturator internus muscle; IG: inferior gemellus muscle; QF: quadratus femoris muscle; LT: lesser trochanter; IT: ischial tuberosity; ST: semitendinosus; BF: biceps femoris; ADM: adductor magnus muscle; SN: sciatic nerve.

**Figure 2 diagnostics-13-00139-f002:**
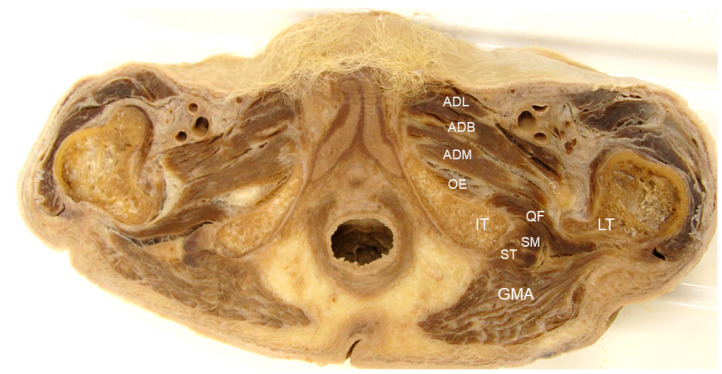
Cadaveric image providing an axial view of the ischiofemoral space. GMA: gluteus maximus muscle; ST: tendon of the semitendinosus; SM: tendon of the semimembranosus; QF: quadratus femoris muscle; OE: obturator externus muscle; LT: lesser trochanter; IT: ischial tuberosity; ADM: adductor magnus muscle; ADB: adductor brevis muscle; ADL: adductor longus muscle.

**Figure 3 diagnostics-13-00139-f003:**
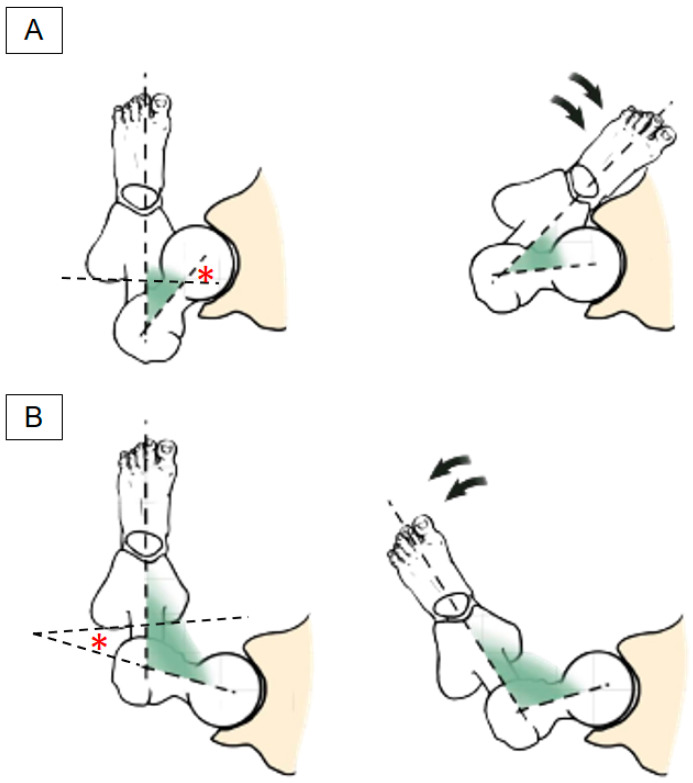
(**A**) Illustration of femoral anteversion (left) and compensatory toe-in posture (right); (**B**) illustration of femoral retroversion (left) and compensatory toe-out posture (right). * denotes the femoral anteversion angle.

**Figure 4 diagnostics-13-00139-f004:**
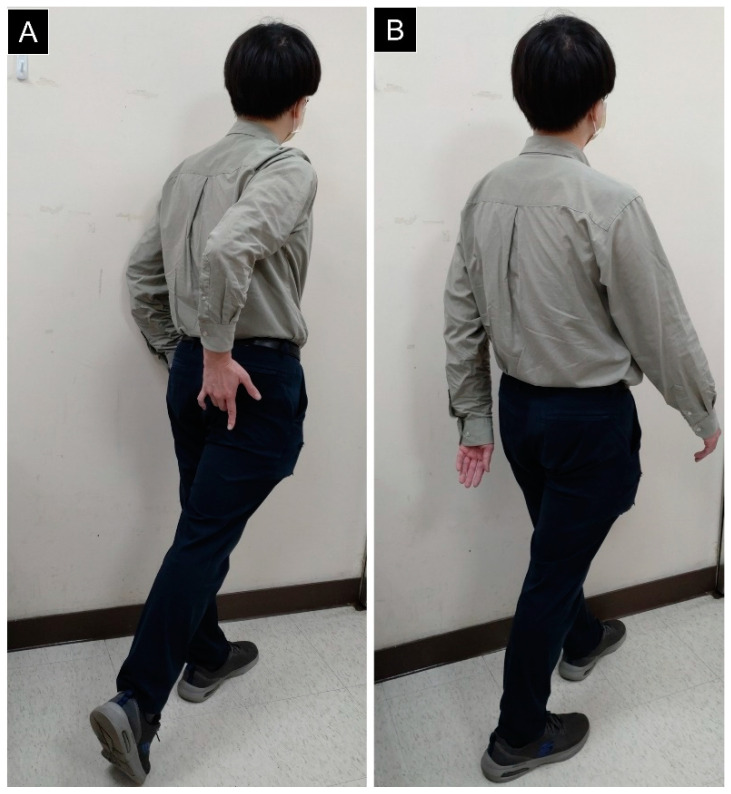
Pain over the ischium can be reproduced by walking with a long stride (**A**) and relieved during short-stride walking (**B**).

**Figure 5 diagnostics-13-00139-f005:**
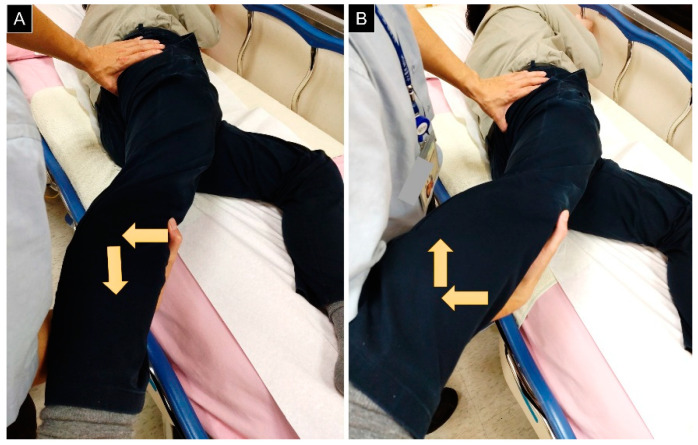
(**A**) Tenderness over the ischium is elicited under passive hip extension and adduction, and it disappears during (**B**) hip extension and abduction. Arrows: direction of force applied by the examiner.

**Figure 6 diagnostics-13-00139-f006:**
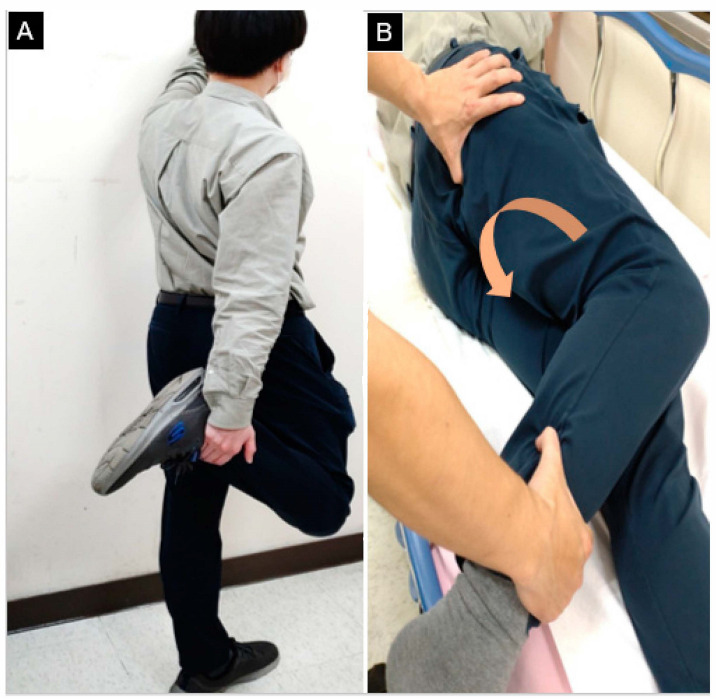
(**A**) Step 1: The volunteer’s hip is brought to extension, external rotation, and adduction to compress the ischiofemoral space in the upright position. (**B**) Step 2: The participant is in the lateral decubitus position with the knee being flexed and the hip being externally rotated (brown arrow). The examiner places the thumb over the ischium to check if the tenderness can be provoked.

**Figure 7 diagnostics-13-00139-f007:**
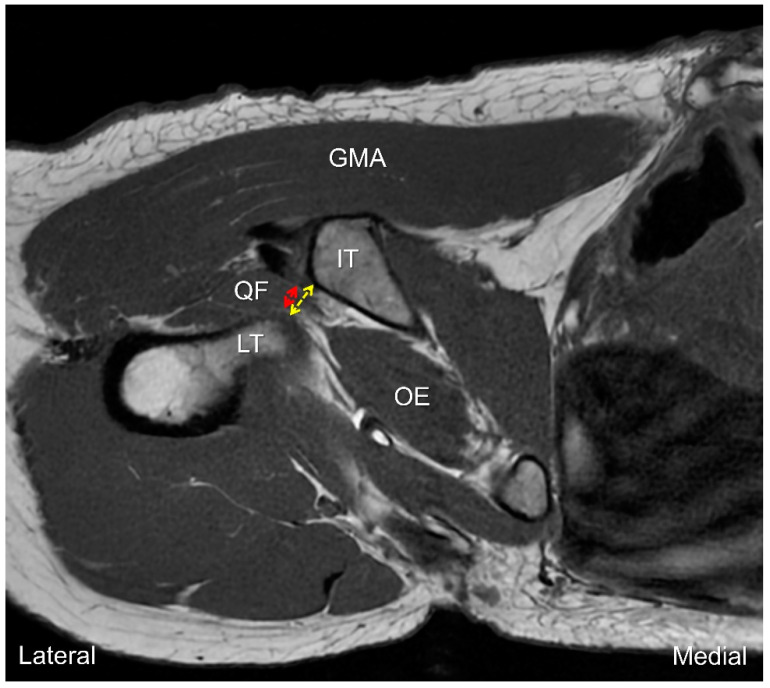
Axial T1 weighted MRI for measurement of the ischiofemoral (yellow dashed line) and quadratus femoris (red dashed line) spaces. GMA: gluteus maximus muscle; QF: quadratus femoris muscle; OE: obturator externus muscle; LT: lesser trochanter; IT: ischial tuberosity.

**Figure 8 diagnostics-13-00139-f008:**
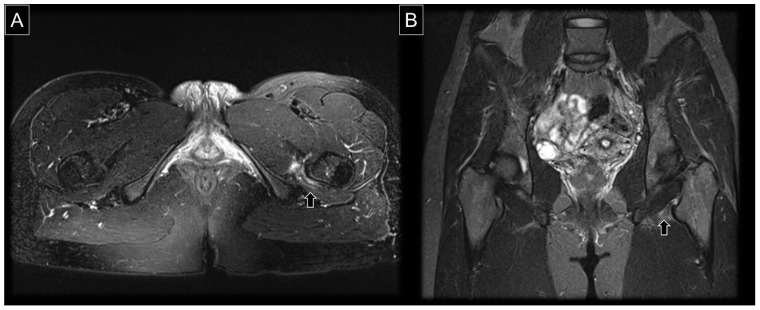
(**A**) Axial and (**B**) coronal views of T2 weighted MRI in a case with IIS. The arrows indicate swelling and perifocal edema of the quadratus femoris muscle.

**Figure 9 diagnostics-13-00139-f009:**
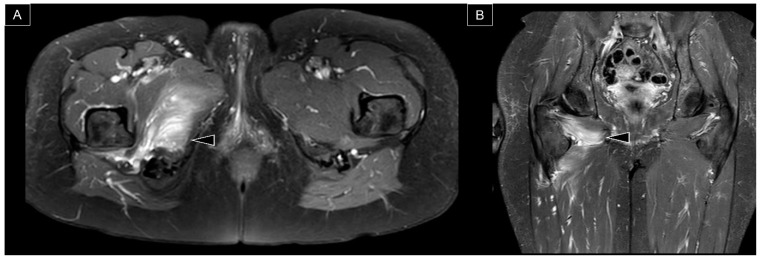
(**A**) Axial and (**B**) coronal views of T2 weighted MRI in another case with post-traumatic IIS, with edema and hematoma inside the quadratus femoris and obturator externus muscles (arrowheads).

**Figure 10 diagnostics-13-00139-f010:**
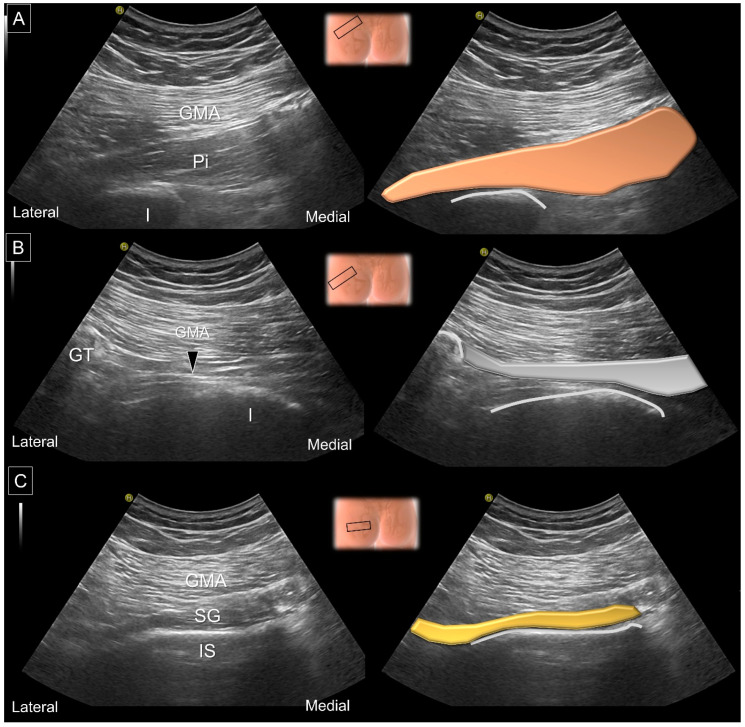
US images and schematic drawings of (**A**) the piriformis muscle and (**B**) its tendon (arrowhead, gray color) and (**C**) the superior gemellus muscle. GMA: gluteus maximus muscle; Pi: piriformis muscle, brown color; SG: superior gemellus muscle, yellow color; IS: ischial spine; I: ischium; GT: greater trochanter. Black hollow square: transducer position.

**Figure 11 diagnostics-13-00139-f011:**
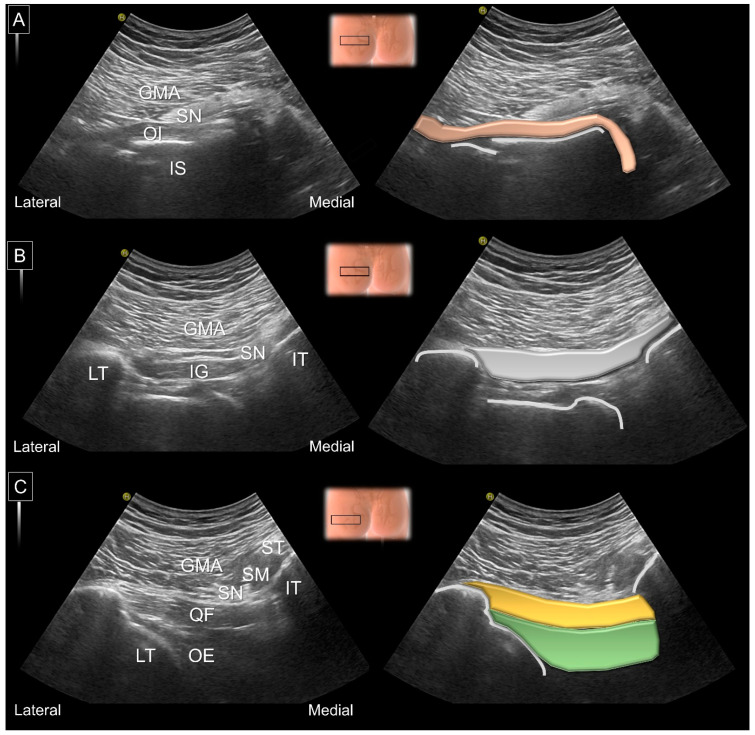
US images and schematic drawings of (**A**) the obturator internus, (**B**) the inferior gemellus and (**C**) the quadratus femoris and obturator externus muscles. GMA: gluteus maximus muscle; IS: ischial spine; OI: obturator internus muscle, light brown color; SN: sciatic nerve; IG: inferior gemellus muscle, gray color; QF: quadratus femoris muscle, yellow color; OE: obturator externus muscle, green shade; LT: lesser trochanter; IT: ischial tuberosity; ST: semitendinosus; SM, semimembranosus muscle.

**Figure 12 diagnostics-13-00139-f012:**
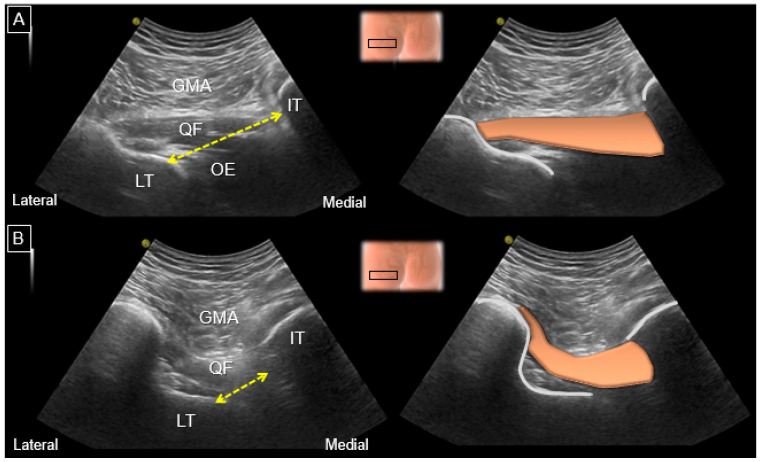
US images and schematic drawings for measuring of the ischiofemoral distance (yellow dashed line) during (**A**) internal and (**B**) external femoral rotations. GMA: gluteus maximus muscle; QF: quadratus femoris muscle, brown color; OE: obturator externus muscle; LT: lesser trochanter; IT: ischial tuberosity.

**Figure 13 diagnostics-13-00139-f013:**
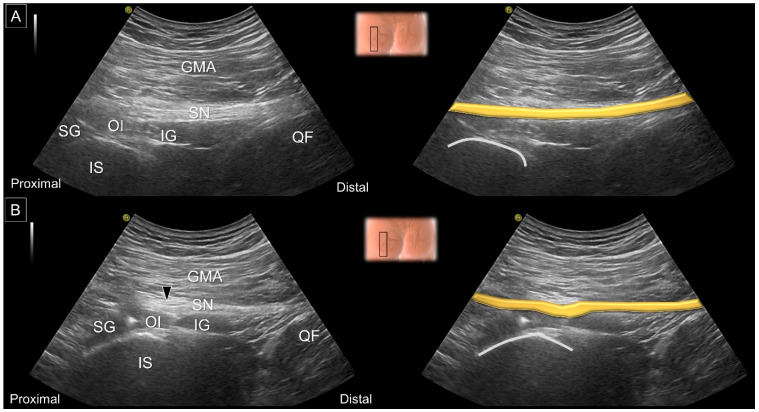
US images and schematic drawings of the sciatic nerve (SN, yellow color) in its long axis in the (**A**) neutral position and (**B**) with hip internal rotation. GMA: gluteus maximus muscle; SG: superior gemellus muscle; IS: ischial spine; OI: obturator internus muscle; IG: inferior gemellus muscle; QF: quadratus femoris muscle; arrowhead, kinking of the sciatic nerve.

**Figure 14 diagnostics-13-00139-f014:**
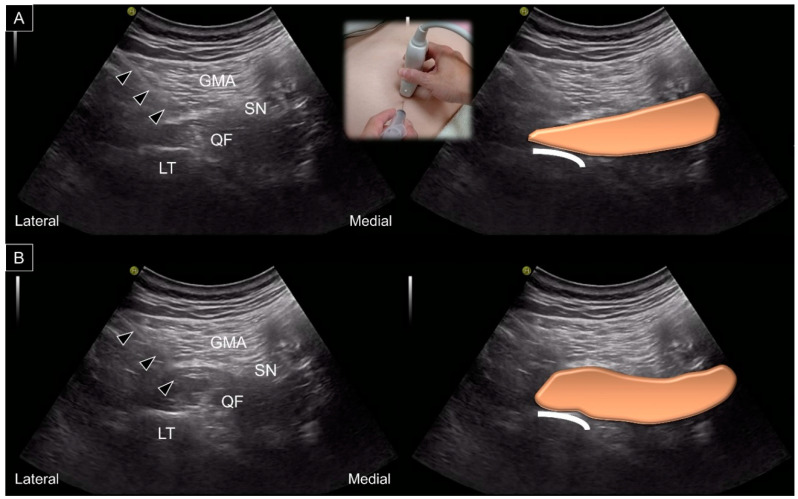
US images and guided injection of the quadratus femoris muscle performed using (**A**) the in-plane lateral-to-medial approach. (**B**) Note the intra-muscular distribution of the injectate. Black arrows: needle trajectory. GMA: gluteus maximus muscle; QF: quadratus femoris muscle, brown shade; LT: lesser trochanter; SN: sciatic nerve.

## Data Availability

Data are contained within the main text of the manuscript.

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
