# Peer review of "Ischiofemoral Impingement Syndrome: Clinical and Imaging/Guidance Issues with Special Focus on Ultrasonography"

_diagnostics, 2022, doi:10.3390/diagnostics13010139_

Round 1
Reviewer 1 Report
Title
Title is appropriate because it is completely informative about the contents of the paper.
Abstract
The abstract respects the rules of the journal. The background and the aim are interesting. In the design is present the type of study. Setting and methods are better explained. The clinical Impact is present and is better explained.
Text
The introduction of the study does clearly sum up the background of the study.
The authors provide a rationale for performing the study based on a review of the medical literature. Furthermore, they define well terms used in the remainder of the manuscript. The hypothesis is defined.
The methods are clear about the methodology.
The results are reported clearly and concisely.
In the discussion, the importance of this syndrome is better emphasized explaining the specific aspects in the daily practice.
Tables
They sum up the study concisely and clearly.
Figures
They sum up the study concisely and clearly.
In figure 11, the image D is missing.
General comments
The purpose of the study is original and real interesting.
Author Response
Comment:
Title
Title is appropriate because it is completely informative about the contents of the paper.
Response:
We appreciate the kind comment from the reviewer.
Comment:
Abstract
The abstract respects the rules of the journal. The background and the aim are interesting. In the design is present the type of study. Setting and methods are better explained. The clinical Impact is present and is better explained.
Response:
We appreciate the kind comment from the reviewer. Regarding settings and methods, we have emphasized this article to be a narrative review. The clinical impact has been highlighted as “Its diagnosis is challenging and requires the combination of physical tests and imaging studies” (line 27-28).
Comment:
Text
The introduction of the study does clearly sum up the background of the study.
The authors provide a rationale for performing the study based on a review of the medical literature. Furthermore, they define well terms used in the remainder of the manuscript. The hypothesis is defined.
The methods are clear about the methodology.
The results are reported clearly and concisely.
In the discussion, the importance of this syndrome is better emphasized explaining the specific aspects in the daily practice.
Response:
We appreciate the kind comment from the reviewer. As this is a narrative review (not an original study), there is no session for discussion. In accordance to the reviewer’s suggestion, we highlighted the importance of this syndrome and its specific aspect in daily practice in the conclusion as “IIS is an overlooked entity within the spectrum of posterior hip pain. It usually presents with symptoms undistinguished from deep gluteal and hamstring syndromes” (line 358-359).
Comment:
Tables
They sum up the study concisely and clearly.
Response:
We appreciate the kind comment from the reviewer.
Comment:
Figures
They sum up the study concisely and clearly.
In figure 11, the image D is missing.
Response: We appreciate the kind reminder from the reviewer. (D) is redundant and is eliminated in the revised manuscript.
General comments
The purpose of the study is original and real interesting.
Response:
We appreciate the kind comment from the reviewer.
Reviewer 2 Report
Dear Authors,
your article is very interesting since it is always hard to correctly address a posterior hip pain in clinical setting.
The way you described the syndrome is very comprehensive and pleasent to read, since every single aspect of the syndrome (from the anatomy to the interventional procedures) was covered.
Thank you for your brilliant contribution and best wishes with any further proceeding related to the manuscript.
Just two minor issues should be fixed:
Line 121; typing mistake:"the"
Line 278: "reliably" instead of reliablility
Author Response
Comment:
Dear Authors,
your article is very interesting since it is always hard to correctly address a posterior hip pain in clinical setting.
The way you described the syndrome is very comprehensive and pleasant to read, since every single aspect of the syndrome (from the anatomy to the interventional procedures) was covered.
Thank you for your brilliant contribution and best wishes with any further proceeding related to the manuscript.
Just two minor issues should be fixed:
Line 121; typing mistake: “the"
Line 278: "reliably" instead of reliablility
Response:
We appreciate the kind comment from the reviewer. In line 121, “the” has been removed from the sentence. In Line 278, the term has been corrected as reliably.